# Patterns of Periodontal Destruction among Smokeless Tobacco Users in a Central Indian Population

**DOI:** 10.3390/healthcare9060744

**Published:** 2021-06-17

**Authors:** Pradeep S. Anand, Supriya Mishra, Deepti Nagle, Namitha P. Kamath, Kavitha P. Kamath, Sukumaran Anil

**Affiliations:** 1Department of Dentistry, ESIC Medical College, Hyderabad 500038, India; dr.pradeep.anand@esic.nic.in; 2Department of Periodontics, Government Dental College and Hospital, Raipur 492001, India; dr.supriya4@gmail.com; 3Department of Periodontics, Faculty of Dentistry, PCDS Campus, People’s University, Bhopal 462037, India; deepti@germanosteo.com; 4Department of Pediatric Dentistry, A. B. Shetty Memorial Institute of Dental Sciences, NITTE (Deemed to be University), Mangalore 575018, India; namithapkamath@nitte.edu.in; 5Private Practice, Specialist Dental Center, Udupi, Mangalore 576104, India; kavithapkamath@yahoo.co.in; 6Department of Dentistry, Oral Health Institute, Hamad Medical Corporation, Doha P.O. Box 3050, Qatar; 7College of Dental Medicine, Qatar University, Doha P.O. Box 2713, Qatar

**Keywords:** gutkha, periodontal conditions, gingival recession, periodontitis, smokeless tobacco, chewers, tobacco, areca nut, smokeless tobacco, oral hygiene

## Abstract

Background: Findings of studies testing the association between smokeless tobacco (SLT) use and periodontal health have shown varying results in different populations. Considering the high prevalence of SLT use in India, the present study was conducted to understand the pattern of periodontal destruction within different areas of the dentition among SLT users. Methods: Age, gender, oral hygiene habits, the frequency and duration of SLT consumption, the type of SLT product used, and the site of retention of the SLT product in the oral cavity were recorded among 90 SLT users. Probing depth (PD), recession (REC), and clinical attachment loss (CAL) at SLT-associated and non SLT-associated teeth of the mandibular arch were compared based on the site of retention of the SLT product, the type of product used, and the duration of the habit. Results: REC and CAL were significantly higher at the SLT-associated zones compared to non SLT-associated zones and at both interproximal and mid-buccal sites of SLT-associated teeth. Among individuals who had the habit for more than 5 years and also among those who had the habit for 5–10 years, PD, REC, and CAL were significantly higher at SLT-associated teeth than at non SLT-associated teeth. Significantly greater periodontal destruction was observed at SLT-associated teeth among khaini users and gutkha users. Conclusions: Smokeless tobacco consumption resulted in greater destruction of periodontal tissues. The severity of periodontal destruction at SLT-associated sites differed depending on the type of smokeless tobacco used, the site of retention of the SLT, and the duration of the habit.

## 1. Introduction

Smoking has been implicated as one of the important factors associated with periodontitis with two to eight-fold increased risk for periodontal attachment and or bone loss [1,2]. Although tobacco smoking seems common worldwide, smokeless tobacco (SLT) is widely used in various forms, such as in betel quid chewing and snuff-dipping or snus habits [3,4]. Smokeless tobacco (SLT) use is highly prevalent in northern and central states of India in the form of betel quid with tobacco, zarda, gutka, khaini, toombak etc., consumed by placing directly in the buccal vestibule or the side of the cheek or lip and chewed [5,6,7,8]. A nationwide survey revealed that the SLT habit is prevalent among 28% of males and 16% of females in the state [8].

In the central Indian state of Madhya Pradesh, where the present study was conducted, SLT consumption is very high and prevalent even among females. The very high prevalence of these habits may be due to a general misconception, particularly among Asian populations, that oral SLT habits are generally less harmful than smoking. Moreover, compared to SLT products available in western countries that are considered to be less harmful than smoking, the products available in Asian countries are more harmful and contain more toxic ingredients [9,10]. Areca nut, classified as Group I human carcinogen, is a major ingredient of products like gutkha and zarda. The common SLT products in India include gutkha and pan masala (powdered tobacco mixed with areca nut, slaked lime, and catechu), betel quid with tobacco, zarda (prepared by boiling pieces of tobacco leaves in water with slaked lime), khaini (tobacco with slaked lime), and mawa (a mixture of areca nut, tobacco, and slaked lime) [11]. Smokeless tobacco products have differences in their composition, preparation, and toxicity and the chemical constituents are nicotine, tobacco-specific N-nitrosamines (TSNA), nitrosamine acids, polycyclic aromatic hydrocarbons (PAHs), aldehydes, and heavy metals [12].

Studies testing the association between SLT use and periodontal health have shown varying results in different populations. Earlier studies conducted in the United States and Sweden have demonstrated that SLT use was associated with increased prevalence of gingival recession [13,14,15,16,17,18,19], while other studies failed to show any association between the use of SLT and the severity of periodontitis [20,21,22,23]. SLT use has been reported to cause increased gingival recession and attachment loss and periodontitis at the adjacent sites [16,24,25,26]. Studies conducted among Asian populations have shown that SLT use is associated with an increased risk for destructive periodontal disease. SLT users in India [27] and Bangladesh [28] have been shown to have an increased risk for tooth loss. Probing depth and attachment loss have been reported to be higher among SLT users when compared to non-users in India [24,25,26,29], Bangladesh [28], and Thailand [30], SLT use was reported to be associated with increased prevalence and severity of gingival recession and attachment loss at mandibular teeth [24] and with higher scores of the community periodontal index, gingival index, and simplified oral hygiene index as well as increased odds for the presence of periodontal disease, pockets, and clinical attachment loss [25,26] among Indian populations. In their study among tobacco users in north India, Singh et al. [31] reported an increased occurrence of recession, attachment loss, mobility, and furcation lesions among SLT users. An epidemiological study conducted among Bangladeshi subjects also showed that SLT use was associated with poor periodontal health as evidenced by increased mean pocket depth, mean attachment loss, and number of missing teeth [28].

Though several studies have reported the deleterious effects of tobacco smoking on the periodontal tissues, only few reports are available on the influence of SLT use on the periodontium. Earlier studies from our group have shown higher prevalence of periodontitis and tooth loss among smokeless tobacco users [24,27,32]. The objective of the present study is to explore the nature of periodontal destruction and its link to the type of SLT and the duration of the habit.

## 2. Materials and Methods

The present study was conducted as a cross-sectional study in the Department of Periodontics, People’s College of Dental Sciences and Research Centre, Bhopal, Madhya Pradesh state, India. The study protocol was approved by the institutional Human Ethics Committee, and written informed consent was obtained from the study participants. SLT users with minimum five teeth remaining in each quadrant who reported for periodontal treatment during January 2012 to December 2012 were enrolled in the study. Individuals who had a history of use of other forms of tobacco (smoking or inhalational), history of use of more than one form of oral SLT product, a history of discontinuation of tobacco use, the presence of any systemic disease, and a history of any form of periodontal treatment or antibiotic therapy during a six-month period prior to the study were excluded from the study. No formal sample size calculation was employed. A convenience sample of individuals who reported to the department and satisfied the study criteria, irrespective of their periodontal status, were used for the purpose of the study. The study protocol was approved by the institutional Human Ethics Committee, and written informed consent was obtained from all of the prospective study participants.

The study variables such as age, gender, oral hygiene habits, frequency, and duration of SLT consumption, type of SLT product used, and site of retention of SLT product in the oral cavity were recorded using a questionnaire. Based on the site of retention of the SLT product in the oral cavity, the participants were categorized into three groups: Group 1—patients who retained the SLT product in the mandibular labial (anterior) vestibule (Zone 1); Group 2—patients who retained the SLT product in the mandibular left buccal vestibule (Zone 2); and Group 3—patients who retained the SLT product in the mandibular right buccal vestibule (Zone 3). SLT users who did not have the habit of retaining the SLT product in any of these regions or who had the habit of retaining it at more than one site were excluded from the study. Based on the duration of the SLT habit, the patients were again grouped into three groups: those who had the habit for <5 years; those who had the habit for 5–10 years; and those who had the habit for >10 years.

Clinical examination was performed to record plaque, calculus, gingival bleeding, probing depth (PD), gingival recession (REC), and clinical attachment loss (CAL). Plaque and calculus scores were recorded on all teeth excluding the third molars in two randomly selected diagonally opposite quadrants. Plaque scores were recorded according to the criteria of the Plaque Index of Sillness and Loe [33], and calculus scores were recorded according to criteria of the calculus component of the Simplified Oral Hygiene Index by Greene and Vermillion [34]. Gingival bleeding was recorded as the presence or absence of gingival bleeding within 30 s after passing the probe through the gingival sulcus on the buccal and lingual surfaces of all teeth [35]. Gingival bleeding score for each individual was recorded as the ratio of the number of sites with gingival bleeding to the number of sites examined [36]. PD, REC, and CAL were recorded at six points on all teeth using a graduated periodontal probe (UNC-15 probe, Hu-Friedy Manufacturing Co., Chicago, IL, USA), and the measurements were rounded off to the nearest millimeter. All clinical recordings were performed by a single trained examiner (SM). Gingival bleeding score for each individual was recorded as the ratio of the number of sites with gingival bleeding to the number of sites examined. PD, REC, and CAL were recorded at six points on all teeth using a graduated periodontal probe (UNC-15 probe, Hu-Friedy Manufacturing Co.), and the measurements were rounded off to the nearest millimeter. All clinical recordings were performed by a single trained examiner (SM).

For calibration purposes, intra-examiner reproducibility was determined by the re-examination of a randomly selected quadrant in 10 patients who were not part of the study. The participants in the calibration exercise were examined twice at 30 min intervals on the same on the same visit by the examiner and intra-class correlation coefficients were determined. Intra-class correlation coefficients for mean PD and mean CAL were 0.916 and 0.902, respectively. Of the replicated measurements, 93.5% were within 1 mm for PD and 91.6% were within 1 mm for CAL.

### Statistical Analysis

Data collected were tabulated and analyzed statistically using SPSS software (SPSS software version 17, IBM Corporation, Armonk, NY, USA). The mean PD, mean REC, and mean CAL were calculated for the buccal and lingual sites of the mandibular teeth. The mean PD, mean REC, and mean CAL of the buccal sites were calculated separately for the mandibular anterior teeth (incisors and canines), mandibular right posterior teeth (right premolars and first and second molars), and mandibular left posterior teeth (left premolars and first and second molars) for each patient. In each patient, mandibular teeth adjacent to the placement site of the tobacco product were designated as SLT-associated teeth and the remaining mandibular teeth were designated as non SLT-associated teeth. The data did not show a normal distribution (Shapiro–Wilk) and were analyzed using non-parametric tests. Comparisons of the periodontal variables at the SLT-associated sites in a particular group with the corresponding sites in the remaining study participants were performed using the Mann–Whitney U-test. Comparisons of mean PD mean REC, and mean CAL between SLT-associated sites and non SLT-associated sites for the whole population among patient groups stratified based on the type of SLT product used and the duration of the habit were performed using the Wilcoxon signed rank test. For all analyses, the statistical significance was fixed at 0.05.

## 3. Results

Among the 90 study participants, 74 were males and 16 females. The age of the participants ranged from 19 years to 70 years, with a mean age of 34.02 ± 12.83 years. The mean scores for plaque, calculus, and gingival bleeding for the study population were 1.31 ± 0.55, 1.52 ± 0.65, and 0.80 ± 0.23, respectively. The mean number of mandibular teeth in the study population was 13.53 ± 1.04. The distribution of the study population in terms of SLT habits is shown in Table 1. Among the 90 study participants, there were 21 in group 1 (participants who retained the SLT product in the mandibular labial vestibule), 39 in group 2 (participants who retained the SLT product in the mandibular left buccal vestibule), and 30 in group 3 (participants who retained the SLT product in the mandibular right buccal vestibule). In terms of the duration of SLT consumption, 44 participants had the habit for <5 years, 26 had the habit for 5–10 years, and 20 had the habit for >10 years. Regarding the SLT product used, 45 consumed gutkha, 20 used khaini, and 25 consumed zarda.

The periodontal parameters at SLT-associated and non-SLT-associated zones are depicted in Table 2. PD was significantly higher at the SLT-associated zones compared to the non-SLT-associated zones for all sites. However, the REC and CAL were significantly higher only for all sites and buccal sites only. Comparison of periodontal parameters at the three different zones of the mandibular arch showed that for the anterior region, PD, REC, and CAL were higher among the SLT-associated teeth. Statistically significant differences were observed for the mean REC for all sites (*p* < 0.05). For zone 2, significantly higher values were observed in group 2 for PD, REC, and CAL at buccal sites. Regarding zone 3, significantly higher values were observed for REC at buccal sites (*p* < 0.05) CAL at all sites (*p* < 0.05) and buccal sites (*p* < 0.05).

Table 3 shows the comparison of periodontal parameters of inter-proximal and mid-buccal sites at the three different zones of the mandibular arch between individuals who retained the SLT product at the zone and individuals who did not retain it at that particular zone. For zone 1, mean REC (*p* < 0.05) and mean CAL (*p* < 0.05) were significantly higher at both the inter-proximal and mid-buccal sites among individuals in group 1 compared to the remaining study population. A similar trend was observed for zones 2 and 3, where individuals who retained the SLT product at these sites exhibited significantly higher mean REC and mean CAL scores at these sites compared to individuals who did not retain the SLT product at these sites. While mean PD was higher at SLT-associated sites in all zones compared to non-SLT-associated sites, statistically significant differences were observed only for inter-proximal sites in zone 3 (*p* < 0.05). Periodontal destruction in terms of mean REC and mean CAL at inter-proximal and mid-buccal sites were more severe in zone 1 than in the remaining areas.

Periodontal variables at buccal sites of mandibular teeth of SLT-associated and non SLT-associated zones for the study population stratified by the duration of SLT use are shown in Table 4. Among individuals who had the habit for <5 years (*n* = 44) and also among individuals who had the habit for 5–10 years (*n* = 26), mean PD (*p* < 0.05), mean REC (*p* < 0.05), and mean CAL (*p* < 0.05 and *p* < 0.001, respectively) were significantly higher at SLT-associated teeth than at non-SLT-associated teeth. However, among individuals who had the SLT habit for >10 years (*n* = 20), although all of the periodontal parameters were higher at SLT-associated teeth than at non-SLT-associated teeth, statistically significant differences were observed only for mean CAL (*p* < 0.05). Individuals who had the habit for >10 years demonstrated more severe periodontal destruction (higher mean PD, mean REC, and mean CAL) even at non-SLT-associated teeth compared to the remainder of the study population.

Table 5 shows the comparison of periodontal variables at buccal sites of mandibular teeth of SLT-associated and non-SLT-associated zones for the study population stratified by the type of SLT product used. Generally, irrespective of the type of SLT product used, mean PD, mean REC, and mean CAL were higher at SLT-associated teeth than at non-SLT-associated teeth. However, statistically significant differences were observed only for mean REC (*p* < 0.05) and mean CAL (*p* < 0.05) among khaini users (*n* = 20) and for mean PD (*p* < 0.001), mean REC (*p* < 0.05), and mean CAL (*p* < 0.001) among gutkha users (*n* = 45).

## 4. Discussion

The present study was performed to determine the pattern of periodontal destruction among smokeless tobacco users in a central Indian population. The results of the present study indicate that the patterns of periodontal destruction among SLT users were significantly different between SLT-associated and non-SLT-associated sites. The findings in the present study suggested that the severity of periodontal destruction at SLT-associated sites varied among individuals depending upon the type of SLT product used, the site of retention of the SLT product, and the duration of the habit. Although studies have suggested that the habit of SLT use may be associated with increased severity of periodontal destruction [25,37], the patterns of periodontal destruction among SLT users are not fully understood. The results reported by different investigators indicate that there are considerable variations in the association of the SLT habit with periodontal disease [19,28]. While a study conducted among Swedish adolescents [19] reported that the prevalence of gingival recession was significantly greater without significant differences in mean CAL among SLT users compared to controls (individuals who never smoked or used snuff), studies conducted among Indian subjects [25] and Bangladeshi subjects [28] have shown that SLT use is associated with increased PD and loss of periodontal attachment. Besides the prevalence of periodontal disease and accessibility to dental care, several factors, such as patterns of SLT habits and types of products, may account for these variations [32].

Nationwide studies conducted in India have shown that smokeless tobacco use is more popular than smoking habit among both males and females in rural as well as in urban areas [8,38,39]. Considering the prevalence of SLT habits among Indians, it is important to understand the harmful effects of such habits on the periodontium. In the present study, it was observed that the probing depth was significantly higher at SLT-associated teeth at the level of the whole tooth, buccal sites, and lingual sites, while significantly higher recession and clinical attachment loss at SLT-associated sites were observed only at the level of the whole tooth and buccal sites. In a similar study conducted among SLT users in a US population, significant differences between SLT-associated and non-SLT-associated teeth were reported only for attachment level at the level of the whole tooth and buccal sites [6]. The participants were categorized into three groups depending on the site of retention of the SLT product in relation to the mandibular arch. The recession and clinical attachment loss were significantly greater at buccal sites among individuals who retained the SLT product at the respective zone. Of the three zones, the maximum recession and clinical attachment loss was observed at the mandibular anterior teeth, suggesting a more severe destruction at these sites. Only few studies reported the periodontal destruction patterns with reference to the site of retention of the SLT product in the oral cavity. These studies have mentioned that the participants retained the SLT product in the lower right or left buccal vestibule [6] or the maxillary anterior tooth region [18,19]. Robertson et al. [16] reported that the majority of the SLT-associated mucosal lesions occurred in the mandibular incisor region with a greater prevalence of gingival recession in these zones. In the present study, most participants retained the SLT product in the mandibular right or left buccal vestibule or the labial vestibule which led to greater periodontal destruction at these sites.

The periodontal parameters at inter-proximal and mid-buccal sites at the three different zones between individuals retained the SLT product at the particular zone showed that both REC and CAL were significantly higher at both inter-proximal and mid-buccal sites. It was also observed that at each zone, recession was higher at mid-buccal sites compared to the inter-proximal sites, whereas for CAL, the trend was reversed. Commonly observed patterns of periodontal disease include deeper probing depths and attachment loss at inter-proximal sites while gingival recession has been more commonly observed to occur at mid-buccal sites. Moreover, among the three zones, higher scores of REC and CAL for both inter-proximal and mid-buccal sites were observed at the mandibular anterior region, suggesting greater periodontal destruction at these teeth. An earlier study comparing periodontal destruction patterns at inter-proximal and mid-buccal sites at SLT-associated sites showed significant differences only for CAL at mid-buccal sites with no significant differences in REC at mid-buccal and inter-proximal sites and CAL at inter-proximal sites [6]. The greater scores of REC and CAL observed at both inter-proximal and mid-buccal sites of SLT-associated teeth in the present study may be due to the difference in the smokeless tobacco habits.

The periodontal parameters such as PD, REC, and CAL at SLT-associated sites were significantly higher among participants who had the habit for less than 10 years. In a similar study among SLT users in the US, significantly greater REC and attachment loss were reported, particularly at buccal sites, at SLT-associated teeth among SLT users who had the habit for more than 10 years, while significant differences were not observed among users who had the habit for less than 10 years [6]. In the present study, all the periodontal parameters of smokeless tobacco users for 10 years or more were higher irrespective of the SLT-associated teeth. This may be because of the long-term use of smokeless tobacco which resulted in generalized changes in the periodontium, a finding similar to that observed in our earlier study among SLT users [24]. The persistence of the habit over a long period of time might have a cumulative effect, which resulted in the generalized damage to the periodontium. This may be explained by the fact that during the habit of tobacco chewing, the harmful ingredients contained in the SLT may be moved around from one region of the oral cavity to the other, thus exposing all areas of the dentition to the deleterious effects of the tobacco contents. Although these products are not retained at all locations for a significant length of time, persistence of the habit over a long period of time may have a cumulative effect, resulting in generalized damage to the periodontium.

The smokeless tobacco products, besides tobacco, include other ingredients such as betel leaf, areca nut, slaked lime, catechu, and spices and the method of preparation also might influence their properties and adverse effects. Hence, it is important to look at the effect of different SLT products on the periodontium. Out of the three types of SLT products studied, zarda resulted in a significantly higher periodontal destruction compared to khaini and gutkha. Gutkha, khaini, and zarda are the most widely used form of smokeless tobacco in India [31,40,41,42]. Though an earlier study by Singh et al. [31] analyzed the severity of periodontal destruction among different types of SLT users, many of the participants in their study used more than one type of product. The SLT products contain more than 4000 different types of mutagenic and carcinogenic ingredients [9,10,43,44]. Nicotine, the principal alkaloid in tobacco, is considered to play a major role in causing periodontal destruction, and its absorption through the oral mucosa is higher from products that have a higher pH and higher unionized nicotine content [44]. In addition to nicotine, these products contain other toxic components that may also contribute to tissue destruction, and this needs to be considered, particularly because an animal study has shown that components other than nicotine in gutkha can exert adverse effects on living tissues [45].

One major limitation of the study is the limited sample size, which was further curtailed when the participants were stratified by the duration of the habit and the type of SLT product used. Moreover, only three types of SLT products were compared in the present study, while at least 10 different types of products are used by different populations in India. As large numbers of people tend to use more than a single product as well as a combination of smoking and SLT, the effects of such habits on the periodontium also need to be evaluated. Another limitation of the study is that we did not do a comparison with SLT and non-SLT users. Influence of oral hygiene practices and etiologic factors such as plaque and calculus also need to be considered in evaluating the role of SLT products on periodontal destruction. Therefore, further studies utilizing larger samples is necessary to understand the adverse effects of smokeless tobacco on the periodontium. Once these associations are better understood, the new information thus generated may be used to better design the public health programs among these populations to tackle periodontal diseases and improve the oral health status among these populations.

## 5. Conclusions

Based on the observations, it can be concluded that periodontal destruction is higher among smokeless tobacco users. The patterns of periodontal destruction showed an association to the area of retention of the product, the duration of the habit, and the type of smokeless tobacco.

## Figures and Tables

**Table 1 healthcare-09-00744-t001:** Distribution of study participants in terms of the site of retention of the SLT product, the duration of SLT use, and the type of SLT product used.

Variables		*N*
Site of retention of SLT	Zone 1-mandibular labial	21
	Zone 2-mandibular left buccal	39
	Zone 3-mandibular right buccal	30
Duration of SLT	<5 years	44
	5–10 years	26
	>10 years	20
Type of SLT	Khaini	20
	Gutkha	45
	Zarda	25

**Table 2 healthcare-09-00744-t002:** Periodontal variables for mandibular teeth at different zones among participants who did (SLT-associated) and did not (non-SLT-associated) retained the SLT product at the zone.

All Zones	Variable	Site	SLT	Non-SLT	*p*-Value
	PD	Whole tooth	3.06 ± 0.81	2.79 ± 0.76	<0.001
		Buccal	3.09 ± 0.81	2.79 ± 0.79	<0.001
		Lingual	3.03 ± 0.96	2.79 ± 0.87	0.003
	REC	Whole tooth	0.78 ± 1.00	0.60 ± 0.96	0.027
		Buccal	1.05 ± 1.25	0.61 ± 1.03	<0.001
		Lingual	0.53 ± 1.00	0.59 ± 1.00	0.322
	CAL	Whole tooth	2.28 ± 2.32	1.70 ± 2.24	<0.001
		Buccal	2.89 ± 2.54	1.72 ± 2.36	<0.001
		Lingual	1.69 ± 2.42	1.69 ± 2.29	0.328
Zone 1	Variable	Site	SLT (*n* = 21)	Non-SLT (*n* = 69)	*p*-value
Anterior	PD	Whole tooth	2.62 ± 0.77	2.51 ± 0.71	0.477
		Buccal	2.83 ± 0.90	2.66 ± 0.80	0.538
		Lingual	2.41 ± 0.75	2.35 ± 0.82	0.525
	REC	Whole tooth	1.35 ± 1.27	0.84 ± 1.28	0.042
		Buccal	1.55 ± 1.39	0.75 ± 1.25	0.004
		Lingual	1.16 ± 1.41	0.94 ± 1.46	0.51
	CAL	Whole tooth	2.79 ± 2.50	1.91 ± 2.52	0.07
		Buccal	3.37 ± 2.44	1.86 ± 2.57	0.009
		Lingual	2.22 ± 2.77	1.97 ± 2.69	0.853
Zone 2	Variable	Site	SLT (*n* = 39)	Non-SLT (*n* = 51)	*p*-value
Left posterior	PD	Whole tooth	3.19 ± 0.76	3.11 ± 0.93	0.186
		Buccal	3.25 ± 0.76	2.99 ± 0.94	0.029
		Lingual	3.14 ± 0.86	3.22 ± 1.03	0.929
	REC	Whole tooth	0.61 ± 0.95	0.34 ± 0.69	0.052
		Buccal	0.83 ± 1.11	0.37 ± 0.82	0.006
		Lingual	0.40 ± 0.88	0.32 ± 0.70	0.853
	CAL	Whole tooth	2.06 ± 2.40	1.49 ± 2.14	0.058
		Buccal	2.59 ± 2.60	1.49 ± 2.23	0.002
		Lingual	1.54 ± 2.38	1.48 ± 2.26	0.621
Zone 3	Variable	Site	SLT (*n* = 21)	Non-SLT (*n* = 69)	*p*-value
Right Posterior	PD	Whole tooth	3.19 ± 0.82	2.94 ± 0.72	0.164
		Buccal	3.03 ± 0.75	2.80 ± 0.78	0.055
		Lingual	3.31 ± 1.05	3.08 ± 0.84	0.387
	REC	Whole tooth	0.61 ± 0.76	0.46 ± 0.86	0.06
		Buccal	0.97 ± 1.24	0.56 ± 1.06	0.018
		Lingual	0.25 ± 0.54	0.32 ± 0.73	0.742
	CAL	Whole tooth	2.24 ± 2.13	1.61 ± 2.22	0.027
		Buccal	2.88 ± 2.55	1.69 ± 2.43	0.007
		Lingual	1.53 ± 2.23	1.49 ± 2.19	0.571

**Table 3 healthcare-09-00744-t003:** Periodontal variables for inter-proximal and mid-buccal sites of mandibular teeth at different zones among participants who did (SLT-associated) and did not (non-SLT-associated) retain the SLT product at the zone.

		Zone 1 (Anterior)			Zone 2 (Left Posterior)			Zone 3 (Right Posterior)		
Site	Variable	SLT (*n* = 21)	Non-SLT (*n* = 69)	*p*-Value	SLT (*n* = 39)	Non-SLT (*n* = 51)	*p*-Value	SLT-Associated (*n* = 30)	Non-SLT (*n* = 60)	*p*-Value
Inter-proximal	PD	3.03 ± 1.00	2.91 ± 0.92	0.735	3.44 ± 0.84	3.18 ± 0.99	0.052	3.34 ± 0.85	3.00 ± 0.83	0.030
	REC	1.44 ± 1.36	0.70 ± 1.23	0.002	0.80 ± 1.09	0.35 ± 0.79	0.017	0.96 ± 1.25	0.59 ± 1.08	0.028
	CAL	3.43 ± 2.46	1.91 ± 2.70	0.009	2.68 ± 2.60	1.56 ± 2.30	0.002	3.11 ± 2.58	1.84 ± 2.52	0.008
Mid-buccal	PD	2.43 ± 0.85	2.17 ± 0.68	0.188	2.87 ± 0.86	2.62 ± 0.96	0.069	2.55 ± 0.82	2.39 ± 0.74	0.397
	REC	1.76 ± 1.50	0.84 ± 1.33	0.004	0.90 ± 1.19	0.39 ± 0.89	0.005	0.99 ± 1.32	0.58 ± 1.04	0.043
	CAL	3.26 ± 2.48	1.77 ± 2.38	0.01	2.41 ± 2.71	1.34 ± 2.15	0.009	2.63 ± 2.60	1.50 ± 2.32	0.025

**Table 4 healthcare-09-00744-t004:** Periodontal variables for mandibular teeth at SLT-associated and non-SLT-associated sites for the study population according to the duration of SLT use.

Duration of Habit	Variable	SLT-Associated	Non-SLT-Associated	*p*-Value
<5 years (*n* = 44)	PD	2.84 ± 0.62	2.54 ± 0.56	0.002
	REC	0.62 ± 0.96	0.38 ± 0.89	0.023
	CAL	1.79 ± 2.05	0.99 ± 1.91	0.003
5–10 years (*n* = 26)	PD	3.08 ± 0.86	2.75 ± 0.80	0.014
	REC	1.04 ± 1.13	0.34 ± 0.61	0.003
	CAL	2.86 ± 2.24	1.22 ± 1.64	<0.001
>10 years (*n* = 20)	PD	3.69 ± 0.84	3.40 ± 0.90	0.070
	REC	1.98 ± 1.50	1.45 ± 1.30	0.121
	CAL	5.36 ± 2.24	3.99 ± 2.71	0.017

**Table 5 healthcare-09-00744-t005:** Periodontal variables for mandibular teeth at SLT-associated and non-SLT-associated sites for the study population according to the type of SLT product used.

Type of SLT	Variable	SLT-Associated	Non-SLT-Associated	*p*-Value
Khaini (*n* = 20)	PD	3.18 ± 0.75	2.90 ± 0.84	0.100
	REC	1.47 ± 1.47	0.75 ± 1.20	0.015
	CAL	3.87 ± 2.38	2.05 ± 2.46	0.002
Gutkha (*n* = 45)	PD	2.89 ± 0.75	2.52 ± 0.59	<0.001
	REC	0.72 ± 1.04	0.28 ± 0.67	0.002
	CAL	1.96 ± 2.21	0.81 ± 1.61	<0.001
Zarda (*n* = 25)	PD	3.39 ± 0.89	3.19 ± 0.89	0.150
	REC	1.30 ± 1.30	1.07 ± 1.22	0.248
	CAL	3.79 ± 2.71	3.11 ± 2.75	0.081

## Data Availability

The datasets generated during and/or analyzed during the current study available from the corresponding author on reasonable request.

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
