# Peer review of "Patterns of Periodontal Destruction among Smokeless Tobacco Users in a Central Indian Population"

_healthcare, 2021, doi:10.3390/healthcare9060744_

Round 1

Reviewer 1 Report

P2 l 52: specify “ very high”

P2 L 69 this is one sentence in stead of two

P3 l 103 I still think that even when a patient took antibiotics more than 6 months, i.e. 8 months, this will still affect the outcome. The washout period should be longer

104 I don’t’ understand why no sample size was calculated

P4 Table 1: please line-out the scores

P7 l 259: habits are….common than smoking; a word is missing

P9 l 337: this is one major drawback of this study, together with the fact that I think it is really hard to be sure the tobacco was held in one exact position over time, in stead of moved through the mouth.

Author Response

  1. P2 l 52: specify “very high”

Modified the text

  1. P2 L 69 this is one sentence instead of two

The sentence modified as “Earlier studies conducted in the United States and Sweden have demonstrated that SLT use was associated with increased prevalence of gingival recession [13-19], while other studies failed to show any association between the use of SLT and the severity of periodontitis [20-23]”.

  1. P3 l 103 I still think that even when a patient took antibiotics more than 6 months, i.e. 8 months, this will still affect the outcome. The washout period should be longer

Agree however in this study we followed the general criteria used in many similar studies conducted in periodontics elsewhere.

  1. 104 I don’t’ understand why no sample size was calculated

As suggested a population-based sample size should have been an ideal choice. However, the time consumed to do all periodontal examination for a single patient was around 30 to 45 minutes. And for the calibration purpose we relied on one examiner doing all measurements. Hence, we were forced to compromise the procedure of taking a community-based sample.

  1. P4 Table 1: please line-out the scores

Followed and the table1 are modified

  1. P7 l 259: habits are….common than smoking; a word is missing

Sentence modified: Nationwide studies conducted in India have shown that smokeless tobacco use is more popular than smoking habit among both males and females in rural as well as in urban areas [8,38,39].

  1. P9 l 337: this is one major drawback of this study, together with the fact that I think it is really hard to be sure the tobacco was held in one exact position over time, in stead of moved through the mouth.

This is true in few states that they keep the quid or the snuff in one location. As mentioned in Kerala , the smokeless tobacco  they move the “quid”  through the mouth.  

Ghosh S, Shukla HS, Mohapatra SC, Shukla PK: Keeping chewing tobacco in the cheek pouch overnight (night quid) increases risk of cheek carcinoma. Eur J Surg Oncol 1996, 22(4):359-360.

Bergstrom J, Keilani H, Lundholm C, Radestad U: Smokeless tobacco (snuff) use and periodontal bone loss. J Clin Periodontol 2006, 33(8):549-554.

Reviewer 2 Report

RE: Patterns of Periodontal Destruction among Smokeless Tobacco Users in a Central Indian Population

  1. L347-348: "Based on the observations it can be concluded that the periodontal destruction among smokeless tobacco users is higher among smokeless tobacco users."  Please check this sentence.
  2. The reviewer believe that comparison of SLYT users with Non-SLT users show more scientific results. Please mention in the limitation.

Author Response

Reviewer 2 : Patterns of Periodontal Destruction among Smokeless Tobacco Users in a Central Indian Population

  1. L347-348: "Based on the observations it can be concluded that the periodontal destruction among smokeless tobacco users is higher among smokeless tobacco users."  Please check this sentence.

Corrected: Based on the observations it can be concluded that the periodontal destruction is higher among smokeless tobacco users.

2. The reviewer believes that comparison of SLYT users with Non-SLT users show more scientific results. Please mention in the limitation.

 It is added to the section on limitations: Another limitation of the study is that we didn’t do a comparison with SLT and Non-SLT users.